# Deep learning empowered sensor fusion boosts infant movement classification

Tomas Kulvicius [1,2,3,12] ✉, Dajie Zhang [2,4,12], Luise Poustka[2], Sven Bölte [5,6,7], Lennart Jahn [1,3], Sarah Flügge[1], Marc Kraft[8], Markus Zweckstetter [2,9,10], Karin Nielsen-Saines [11], Florentin Wörgötter [3,13] & Peter B. Marschik [1,2,4,5,13] ✉

## Abstract

**Background** To assess the integrity of the developing nervous system, the Prechtl general movement assessment (GMA) is recognized for its clinical value in diagnosing neurological impairments in early infancy. GMA has been increasingly augmented through machine learning approaches intending to scale-up its application, circumvent costs in the training of human assessors and further standardize classification of spontaneous motor patterns. Available deep learning tools, all of which are based on single sensor modalities, are however still considerably inferior to that of well-trained human assessors. These approaches are hardly comparable as all models are designed, trained and evaluated on proprietary/silo-data sets.

**Methods** With this study we propose a sensor fusion approach for assessing fidgety movements (FMs). FMs were recorded from 51 typically developing participants. We compared three different sensor modalities (pressure, inertial, and visual sensors). Various combinations and two sensor fusion approaches (late and early fusion) for infant movement classification were tested to evaluate whether a multi-sensor system outperforms single modality assessments. Convolutional neural network (CNN) architectures were used to classify movement patterns.

**Results** The performance of the three-sensor fusion (classification accuracy of 94.5%) is significantly higher than that of any single modality evaluated.

**Conclusions** We show that the sensor fusion approach is a promising avenue for automated classification of infant motor patterns. The development of a robust sensor fusion system may significantly enhance AI-based early recognition of neurofunctions, ultimately facilitating automated early detection of neurodevelopmental conditions.

## Plain language summary

Study of the movements of infants enables evaluation of development. We explored whether combining information obtained from different types of detectors, able to assess pressure, motion, and visually, improved the accuracy of results. Different ways to combine data from these different detectors were tested, and it was found that using all three together produced the most accurate results. Our approach could be further developed to allow more reliable automated tools to detect problems with development in infants, potentially leading to earlier diagnosis and intervention in disorders such as cerebral palsy.

In recent years, we have seen a boom in the development of automated solutions for the Prechtl general movements assessment (GMA). These efforts utilize AI methods in the attempt to improve classic clinical assessments by removing potential subjectivity in implementation and reducing the long term costs associated with training and qualifying human experts[1–7]. The original GMA, introduced in the late 1990s, is a validated diagnostic tool based on a human gestalt appraisal of infants' endogenously generated motor functions to detect neurological impairments within the first few months of life[8]. Classic GMA has become, alongside MRI and the Hammersmith Infant Neurological Examination, HINE, a gold standard diagnostic tool for the early detection and prediction of cerebral palsy, CP[9]. Aside from detecting CP early, GMA has shown to identify prodromal

motor abnormalities in neurodevelopmental conditions like autism spectrum disorder (ASD), genetic disorders like Rett syndrome (RTT), and disorders related to mothers' viral infections during pregnancy, e.g., Zika virus, Sars-CoV-2, HIV[10–16]. Compared to other diagnostic tools such as MRI, GMA can be implemented by experts within minutes without the need of anyone touching the infant, while delivering unbeatable diagnostic accuracy[8,9]. GMA, however, can only be performed by well-trained and certified assessors. Access to qualified training is not always available. Also, assessors need continuous practice and regular re-calibrations to ensure high performance. Training and re-calibrating assessors are costly in the long-run. Although assessors' reliability has proven excellent across various sites and studies[17–20], human and environmental factors will remain an issue

---

that could affect individual performance[2]. In regions and remote settings with no experts on-site, applying GMA is still challenging. These are the main reasons why GMA has not yet been globally established in daily clinical routines. An ever-growing search for complementing avenues to scale up GMA consequentially arose, putting technological advancements at the forefront for method improvement and applied clinical research. Recent automated GMA attempts mainly focus on identifying a single medical condition by tracking and classifying a fraction of motor patterns which the original GMA evaluates. We are still a long way away for the proposed AI methodology to approach the utility of the original GMA, both in regards to technology and diagnostic scope.

Different sensor modalities for AI-driven GMA have been developed, each with its own strengths and limitations[1–7,21]. Most of these automated approaches were developed based on visual data with RGB/RGB-D imaging sensors[7,22–32], which is also the data source of 'original GMA'. Methods using marker-based body tracking[33] or wearable sensors, e.g., inertial measurement units (IMUs)[34–41], encouraged by their success in adult motion tracking, were also proposed for use in infants. Pressure sensing devices have been employed in the assessment of general movements[5,21,42,43], since they are non-intrusive and easy to use, hence being particularly suitable for clinical practice. At present, the development of automated GMA approaches is still in its infancy, mostly based on small datasets, with minimal data-sharing or pooling, and targeting only a fraction or a specific aspect of the tasks involved in the standard GMA. Most AI methods focus understandably on the classification of fidgety movements (e.g., present or absent, typical or atypical[1–7,21,29], a general movements pattern of high diagnostic value for the early detection of neurodevelopmental integrity at around three months post-term age[8,9,44]. The classification performances of single modality methods, irrespective of their task specifics and different participants samples, compete with each other with an accuracy level of about 90%. Different methodologies cannot be compared to each other as all models are designed, trained and evaluated on proprietary datasets[1–7,21,29,38,45–47].

Targeting a higher accuracy, we proposed a multi-sensory recording setup and sensor fusion approach for automated GMA, utilizing different combinations of video cameras, pressure sensing device, and IMUs[5,37,38,46–48]. The rationale behind the sensor fusion approach is that each motion tracking modality captures different types and dimensions of movement information (e.g., position and amplitude in space, body parts involved, force, velocity, frequency, direction, angular velocity and acceleration, etc.) which other sensors may miss or are unable to track directly. If different

inputs are integrated, they may compensate and boost each other towards better performance. This approach needs to be empirically tested. To the best of our knowledge, there exists no public-accessible multi-sensory dataset of general movements, while no study has carried out any empirical comparisons of utilities of different sensing modalities, let alone their combinations, for analyzing infant motor functions and enhancing classification accuracy.

To fill in this knowledge gap, we perform classification experiments using convolutional neural network (CNN) architectures, and different sensor modalities and their combinations to address three research questions: (1) Do performances of different sensor modalities differ from each other for the same task (i.e., tracking and classifying fidgety vs. non-fidgety movements)? (2) Does sensor fusion outperform single modality assessments and lead to higher accuracy in infant movement classification? (3) Is a sensor fusion approach with non-intrusive sensors sufficient for accurate movement tracking and classification?

The main contributions and outcomes of this work are as follows. We present and share a labeled and fully synchronized multi-sensory (pressure, inertial, and visual) dataset of infant movements. We propose a sensor fusion approach and undertake a deep learning-empowered comparison of these three sensor modalities and their various combinations for movement classification. We provide evidence that multi-sensory approach has great potential to further improve movement classification. We compare two different sensor fusion approaches, early (using one neural network) vs. late (combining outputs of multiple neural networks) sensor fusion, and discuss their value for the automated classification of motor patterns. This work informs future endeavors utilizing AI methods with respect to clinical evaluation of movements such as spontaneous general movements in infants at high risk for adverse neurological outcomes.

## Methods
### Method overview
A flow diagram of the study pipeline is shown in Fig. 1. In this study we used data collected and pre-processed by our research group at iDN's BRAINtegrity lab, Medical University of Graz, Austria, and Systemic Ethology and Developmental Science (SEE) Labs, University Medical Center Göttingen and Heidelberg University, Germany[29,46]. We collected and segmented 19451 data units from 51 biweekly assessed participants[21,29]. The study aimed to analyze the ontogeny of human behavior and typical cross-domain development during the first months of human life[46]. The project was

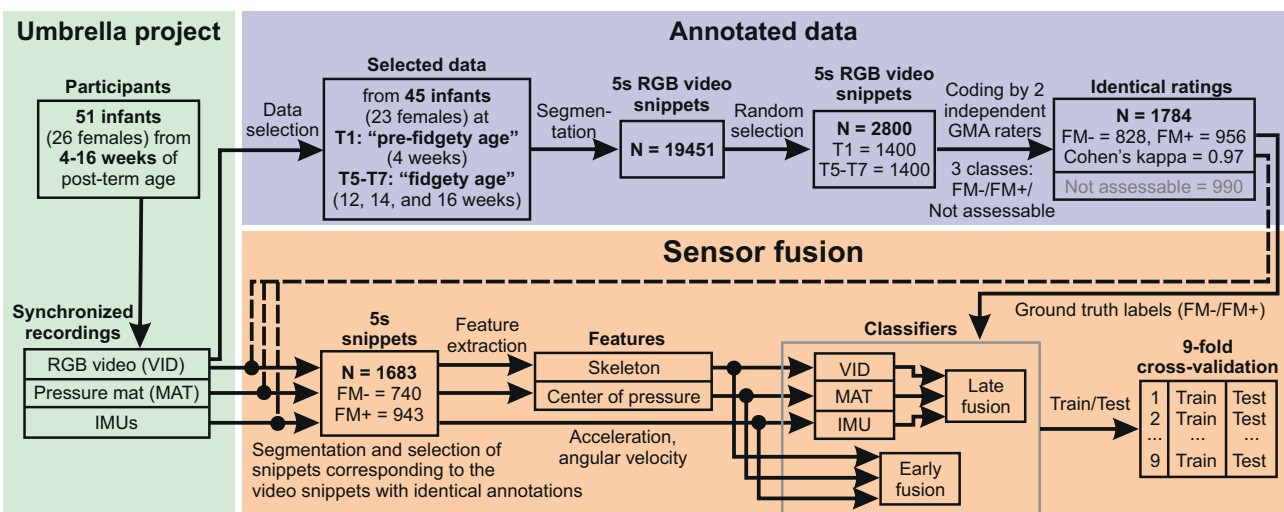

**Fig. 1 | Flow diagram of the study pipeline.** N corresponds to the number of snippets (5 s data units) in each step. T1-T7 correspond to seven recording sessions in biweekly intervals, starting at 4 weeks post-term age. FM- and FM+ corresponds to the absence and presence of fidgety movements, respectively. VID -- video data, MAT -- pressure mat data, IMU -- inertial measurement unit data.

approved by the Institutional Review Boards of the Medical University of Graz, Austria (27-476ex14/15) and the University Medical Center Göttingen, Germany (20/9/19).

In this study, we adopted movement data captured by three different sensor modalities: a 2D RGB camera, a pressure sensing mat, and six inertial measurement units (IMUs). For movement classification based on video data, we used skeleton features as presented in prior studies[29,45]. For movement classification based on pressure mat data, we used center of pressure (CoP) features as presented in a prior study[21]. Data from the IMUs and IMU-based classification are reported here for the first time. Based on our previous work, for the movement classification we used convolutional neural network (CNN) architectures, which is superior for the current task compared to other classification methods such as multi-layer perceptron (MLP), support vector machine (SVM) and long short-term memory (LSTM) network[21].

We compared the performance of movement classification when (a) using single sensor modalities, (b) using two-sensor fusion (combinations of two different sensor modalities), and (c) using a three-sensor fusion model (a combination of all sensor modalities). For the evaluation and comparison of these approaches, we used a 9-fold cross-validation procedure where test sets contained data only from infants not included in the training sets.

In the following sections, we provide details on data acquisition and processing, movement analysis, and classification.

## Dataset

**Participants**. 51 infants born between 2015 and 2017 from monolingual German-speaking families were sampled. All parents of the participating typically developing (TD) infants provided written informed consent to study participation and publication of depersonalized data. Infant inclusion criteria were: uneventful pregnancy, uneventful delivery at term age (>37 weeks of gestation), singleton birth, appropriate birth weight, uneventful neonatal period, inconspicuous hearing and visual development. All parents completed high-school or higher level of education and had no record of alcohol or substance abuse.

We post-hoc excluded one infant due to a rare genetic disorder diagnosed at three years of age. Another five infants were excluded due to the lack of full recordings within the required age periods. Thus, the final sample for this study consisted of 45 infants (23 females).

**Multi-modal movement recordings**. We recorded infant movements from 4 to 16 weeks of post-term age in biweekly intervals at 7 succeeding sessions in a standardized laboratory setting. Data recording procedure followed the standard GMA guidelines[49]. Time-points for the seven sessions were T1: 28 ± 2 days, T2: 42 ± 2 days, T3: 56 ± 2 days, T4: 70 ± 2 days, T5: 84 ± 2 days, T6: 98 ± 2 days, and T7: 112 ± 2 days corrected age.

According to the GMA manual[49], 5 to 8 weeks of post-term age mark a period (periods T2 and T3) of transitional movements which is considered a "grey"-zone between the writhing and the fidgety movement (FM) periods, which is not ideal for assessing infant general movements. FMs are most pronounced in typically developing infants from 12 weeks of post-term age on-wards (corresponding to the T5-T7 periods)[49]. Therefore, to analyze infant general movements, recordings from T1 as the "pre-fidgety period" and T5-T7 as the "fidgety period" were used. Each session, infants were dressed with specifically designed body-suits and placed in a supine position in a standard crib by the parent. We recorded infant movements using an RGB camera, a pressure sensing mat, and six inertial measurement units (IMU). All sensor recordings were synchronized[46].

For video recordings (see Fig. 2a), we used a standard HD camcorder mounted on an aluminum-frame affixed to the cot (please also see ref.[46]). The HD camcorder had a resolution of 1920 × 1080 pixels, and a frame rate of 50 frames per second. Note that nowadays camcorder could be replaced by a smartphone with a camera of similar resolution and frame rate (e.g., see[22,50]). The pressure data was acquired using a Conformat pressure sensing mat (Tekscan, Inc., South Boston, Massachusetts, USA[51]). This pressure sensing mat contains 1024 pressure sensors arranged in a 32 × 32 grid array on an area of 471.4 × 471.4 $mm^2$. The mat was laid on the crib mattress and covered by a cotton sheet. The pressure mat produced pressure image frames (see Fig. 2b) with a resolution of 32 × 32 pixels, and 100 $Hz$ sampling rate[21]. To capture infant motion using wearable sensors, six wireless Xsense MTw Awinda IMUs[52] were applied. Each IMU sensor contains a 3-axis accelerometer and a 3-axis gyrometer, which measure acceleration and angular velocity in $X$, $Y$, and $Z$ directions with a sampling rate of 60 $Hz$. Four IMU sensors were each fed into a designated pocket of a customized body-suit and attached to the infant's shoulders and the outsides of the hips to assess proximal movement features. For each session, a body-suit in appropriate size for the infant was worn. The other two IMUs were each put into a sock and attached to the soles of the infant's feet for distal features in the lower extremities (for sensor locations see Fig. 2c). Data was synchronized using time stamps for each sensor modality. Data were then aligned (synchronized) based on these time stamps.

**Movement annotations**. For this study, to train and test classifiers, we used human-annotation data (see Fig. 1). The coders were two senior GMA experts with continuous and more than 20 years practice. Annotations were performed on video recordings, which were synchronized with IMU sensors and the pressure sensor such that the same labels were used for all sensor modalities. To annotate video data, videos were first cut such that infants were overall awake and active, and not fussy. We determined the length of video snippet to be 5 seconds, as a minimum length of the video for human assessors feeling confident to judge whether the fidgety movement is present (FM+) or absent (FM-)[21,29].

In a proof-of-concept study[29], a fraction of the total available snippets (N = 19451) was randomly sampled (N = 2800) and annotated by two experienced GMA assessors: 1400 from T1, the pre-fidgety period, and 1400 from T5-T7, the fidgety period (see Fig. 1). Both assessors, blinded in regards to infant ages, evaluated each of the 2800 randomly ordered snippets (5 s long) independently, labeling each snippet as "FM+", "FM-", or "not assessable" (i.e., in case the infant was fussy, crying, drowsy, hiccuping, yawning, refluxing, exhibiting a pleasure burst, self-soothing, or distracted for the respective 5 s, all of which can distort an infant's movement pattern rendering the recording inadequate for GMA[8,49]).

Cohen's kappa for the interrater agreement of the two assessors for classes FM+ and FM- was κ = 0.97, whereas the intrarater agreement by re-rating 10% randomly selected snippets was κ = 0.85 for assessor 1, and κ = 0.95 for assessor 2. Snippets with non-matching FM+/FM- labels between two assessors (N = 26) and the ones labeled as "not assessable" by either assessor (N = 990) were excluded for further analysis. Thus, remaining 1784 video snippets were labeled identically by both assessors as either FM+ (N = 956) or FM- (N = 828). Out of the 1784 video snippets, 101 had no corresponding synchronized IMU and/or pressure mat data (due to synchronization issues), therefore, in this study, for the classification experiments 1683 (943 FM+ and 740 FM-) snippets were used.

## Data pre-processing and feature extraction

**Video data**. For video-based movement classification, we used body key points as features (Fig. 2a; see also refs.[29,45]). In this study, we extracted body key points using the state-of-the-art pose estimation framework ViTPose[53], which is more accurate than OpenPose[54,55] that we used in our previous studies[29,45]. ViTPose extracts 17 body key points as shown in Fig. 2a including key points on both ears (not shown). We excluded the two ear key points since three head key points (nose and eyes) are sufficient to determine the position and orientation of the head. Thus, 15 key points were then used throughout for analyses. Given a video snippet

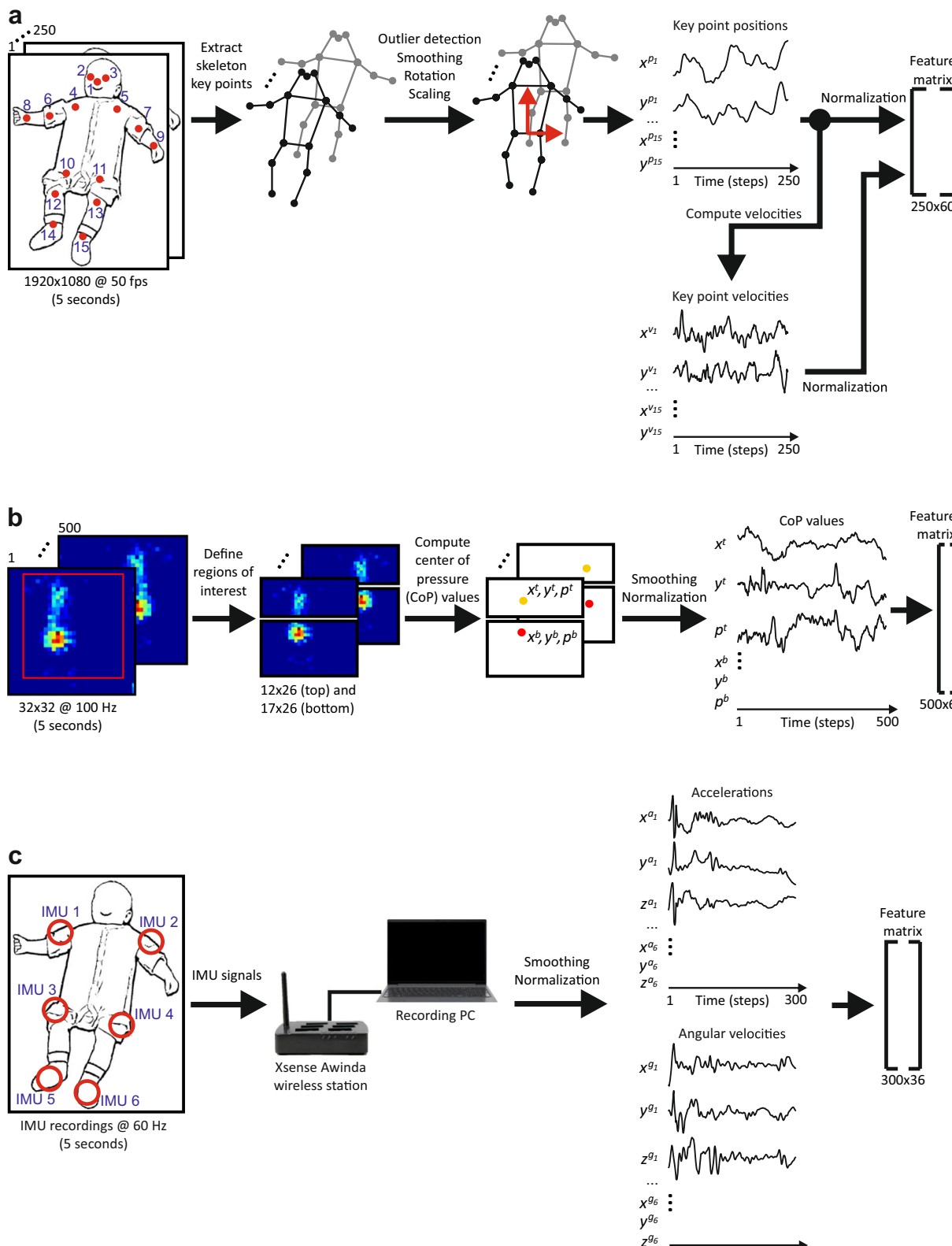

**Fig. 2 | Flow diagrams of the feature extraction procedures for three sensor modalities. a** video data, **b** pressure mat data, and **c** IMU sensor data.

length of 5 *s*, and frame rate of 50 *Hz*, we obtained 250 frames per video snippet with 15 key points per frame.

Several pre-processing steps were performed such as detection and removing of outliers, smoothing, centering, rotation and scaling of the skeleton key points, and normalization of position and velocity values.

We denote position values for $X$ and $Y$ coordinates of the key points as $x^p$ and $y^p$. First, to remove outliers (i.e., key points that were detected incorrectly) and to smooth the movement trajectories, we applied median and moving average filters with a sliding window size of 5 frames.

Then we centered body key points with respect to the average center point between the hip key points ($h^c$) across all frames by:

$$h^c = \left[\frac{1}{2n}\left(\sum_f x_f^{p_{10}} + \sum_f x_f^{p_{11}}\right), \frac{1}{2n}\left(\sum_f y_f^{p_{10}} + \sum_f y_f^{p_{11}}\right)\right],$$

$$x_f^{p_k} \leftarrow x_f^{p_k} - h_1^c, y_f^{p_k} \leftarrow y_f^{p_k} - h_2^c, \tag{1}$$

where $k = 1\ldots15$ is the key point index, and $f = 1\ldots n$ ($n = 250$) is the frame index. $x_f^{p_{10,11}}$ and $y_f^{p_{10,11}}$ correspond to the $X$ and $Y$ coordinates of the hip key points (see Fig. 2a).

Afterwards, for each snippet we rotated the skeletons such that the middle line between the average hip and shoulder key points is aligned with the $Y$ axis (see Fig. 2a):

$$s^c = \left[\frac{1}{2n}\left(\sum_f x_f^{p_4} + \sum_f x_f^{p_5}\right), \frac{1}{2n}\left(\sum_f y_f^{p_4} + \sum_f y_f^{p_5}\right)\right],$$

$$\alpha = acos\left(s^c \cdot [0\ 1]^T / ||s^c||\right), R = \begin{bmatrix} cos(\alpha) & -sin(\alpha) \\ sin(\alpha) & cos(\alpha) \end{bmatrix},$$

$$\begin{bmatrix} x_f^{p_k} \\ y_f^{p_k} \end{bmatrix} \leftarrow R \cdot \begin{bmatrix} x_f^{p_k} \\ y_f^{p_k} \end{bmatrix}, \tag{2}$$

where $s^c$ is the average center point between shoulder key points, $\alpha$ is the rotation angle, and $R$ is the rotation matrix. $x_f^{p_{4,5}}$ and $y_f^{p_{4,5}}$ correspond to the $X$ and $Y$ coordinates of the shoulder key points (see Fig. 2a).

Next, for each snippet we scaled size of the skeletons such that the body length between the hip and shoulder center points ($h^c$ and $s^c$) is equal to 1/3:

$$x_f^{p_k} \leftarrow \frac{x_f^{p_k}}{3|h_2^c - s_2^c|}, y_f^{p_k} \leftarrow \frac{y_f^{p_k}}{3|h_2^c - s_2^c|}. \tag{3}$$

Finally, for each snippet we centered the time series of each key point by subtracting the mean value of the corresponding time series.

In addition to the positions of the key points ($x_f^{p_k}, x_f^{p_k}$), we also used their velocities ($x_f^{v_k}, x_f^{v_k}$). Since the scale of velocity values is much smaller than the scale of position values, we normalized position and velocity values separately using z-score normalization. For this, we calculated mean and standard deviation values across all position (or velocity) time series in the training sets and then used these values to normalize the data in the training and test sets.

After pre-processing, and concatenation of position and velocity features, this led to the feature matrix of size $250 \times 60$:

$$\begin{bmatrix} x_1^{p_1} & y_1^{p_1} & \ldots & x_1^{p_{15}} & y_1^{p_{15}} & x_1^{v_1} & y_1^{v_1} & \ldots & x_1^{v_{15}} & y_1^{v_{15}} \\ x_2^{p_1} & y_2^{p_1} & \ldots & x_2^{p_{15}} & y_2^{p_{15}} & x_2^{v_1} & y_2^{v_1} & \ldots & x_2^{v_{15}} & y_2^{v_{15}} \\ \vdots & \vdots & \ddots & \vdots & \vdots & \vdots & \vdots & \ddots & \vdots & \vdots \\ x_n^{p_1} & y_n^{p_1} & \ldots & x_n^{p_{15}} & y_n^{p_{15}} & x_n^{v_1} & y_n^{v_1} & \ldots & x_n^{v_{15}} & y_n^{v_{15}} \end{bmatrix}, \tag{4}$$

with $n = 250$ frames.

Note that we also tried to use accelerations of the key points, however, preliminary analysis showed that including accelerations did not improve classification accuracy (see Supplementary Fig. 1). Therefore, we did not include accelerations for further analysis.

**Pressure mat data**. The pre-processing and feature extraction procedure of the pressure mat data is shown in Fig. 2b. We used synchronized 5

$s$ snippets corresponding to the video snippets which hence bear the same labels (FM+/FM-). Given a sampling rate of 100 $Hz$ and the sensor grid resolution of $32 \times 32$ sensors, resulting in 500 frames per snippet, and 1024 pressure values per frame. As features, we used center of pressure (CoP) coordinates $x^{t/b}, y^{t/b}$, and average pressure value $p^{t/b}$ of the top and bottom areas[21] as shown in Fig. 2b.

We first cropped the area of size $29 \times 26$ ([1:29, 4:29]) of the original grid size (see red rectangle in Fig. 2b), since the sensor values outside this area were 0 in most of the cases. The cropped area contained 754 pressure sensor values. Generally, only two areas were strongly activated on the pressure mat (see Fig. 2b). The activation at the top corresponds to the infants' shoulders and/or head, whereas activation at the bottom corresponds to the infant's hips. Thus, we split the cropped grid area of size $29 \times 26$ into two parts, $12 \times 26$ (top part) and $17 \times 26$ (bottom part), and tracked the center of pressure (CoP) in these two areas.

Next, we computed position coordinates $x^{t/b}$ and $y^{t/b}$ of the CoP and the average pressure values $p^{t/b}$ of the top ($t$) and the bottom ($b$) areas for each frame (we skip frame index for brevity):

$$x^{t/b} = \frac{\sum_{i,j} j \times p^{t/b}(i,j)}{\sum_{i,j} p^{t/b}(i,j)}, \ y^{t/b} = \frac{\sum_{i,j} i \times p^{t/b}(i,j)}{\sum_{i,j} p^{t/b}(i,j)}, \ p^{t/b} = \frac{\sum_{i,j} p^{t/b}(i,j)}{m^{t/b} \times n^{t/b}}, \tag{5}$$

where $p^t(i, j)$ and $p^b(i, j)$ correspond to the pressure sensor values at the position $i, j$ ($i = 1\ldots m^{t/b}, j = 1\ldots n^{t/b}$) of the top and bottom areas, respectively.

To reduce signal noise, for each time series $x^{t/b}, y^{t/b}$, and $p^{t/b}$, we applied the moving average filter with a sliding window size of 5 frames. To avoid biases that could be caused by infant's body size and weight, we normalized position and pressure values across top and bottom areas for each snippet between 0 and 1 as follows:

$$\gamma = \max\left(\left[\max(x^t) - \min(x^t), \ \max(y^t) - \min(y^t),\right.\right.$$
$$\left.\left. \max(x^b) - \min(x^b), \ \max(y^b) - \min(y^b)\right]\right),$$

$$x^{t/b} \leftarrow \frac{x^{t/b} - \min(x^{t/b})}{\gamma}, \ y^{t/b} \leftarrow \frac{y^{t/b} - \min(y^{t/b})}{\gamma}, \tag{6}$$

$$\beta = \max([\max(p^t) - \min(p^t), \max(p^b) - \min(p^b)]),$$
$$p^{t/b} \leftarrow \frac{p^{t/b} - \min(p^{t/b})}{\beta}. \tag{7}$$

Finally, this led to the feature matrix of size $500 \times 6$:

$$\begin{bmatrix} x_1^t & y_1^t & p_1^t & x_1^b & y_1^b & p_1^b \\ x_2^t & y_2^t & p_2^t & x_2^b & y_2^b & p_2^b \\ \vdots & \vdots & \vdots & \vdots & \vdots & \vdots \\ x_n^t & y_n^t & p_n^t & x_n^b & y_n^b & p_n^b \end{bmatrix}, \tag{8}$$

with $n = 500$ frames.

Note that in addition to $x^{t/b}, y^{t/b}$, and $p^{t/b}$ features we also tried to use their first derivatives, however, preliminary analysis showed that including derivatives as additional features did not improve classification accuracy (see Supplementary Fig. 2). Therefore, we did not include derivatives for further analysis.

**Inertial measurement unit data**. Similar to the pressure mat data, we used synchronized 5 $s$ snippets corresponding to the video snippets. Given an IMU sampling rate of 60 $Hz$, this led to 300 frames per snippet. For each IMU sensor, we obtained three accelerometer values and three gyrometer values (for each coordinate $X$, $Y$, and $Z$), thus, in total 36 values per frame (see Fig. 2c).

**Fig. 3 | Classification models. a** Schematic diagram of the convolutional neural network (CNN) architecture. Hyperparameters for different sensor modalities are specified in Supplementary Tables 2 and 3. Schematic diagrams for two sensor fusion approaches: combination of three networks trained using single sensor modalities – late sensor fusion (**b**), and one network trained using all sensor modalities – early sensor fusion (**c**).

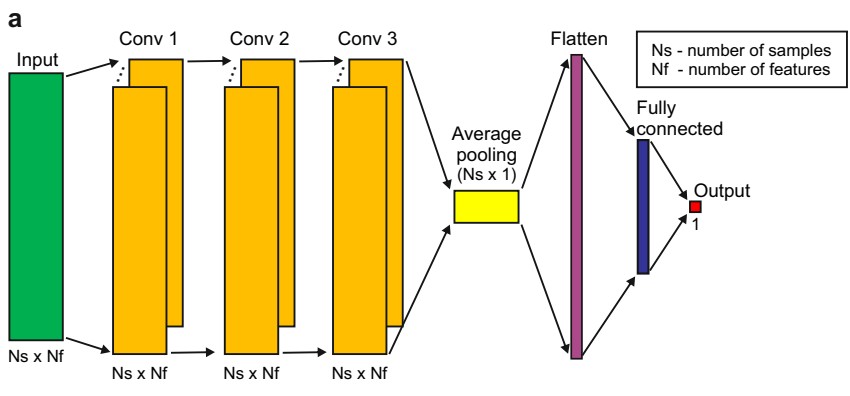

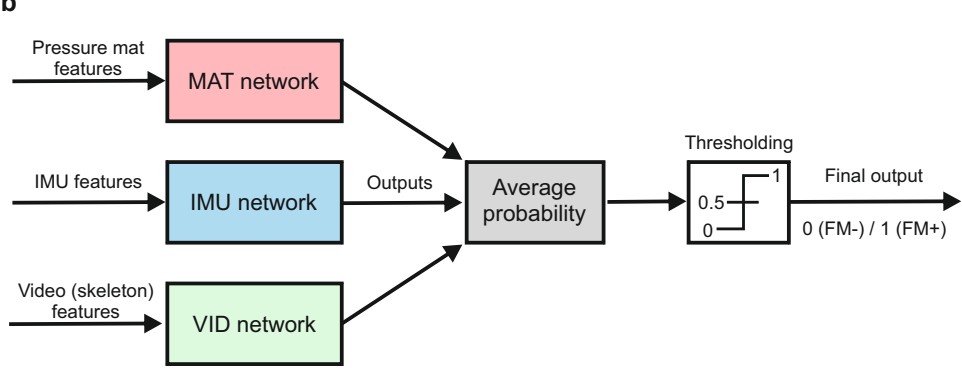

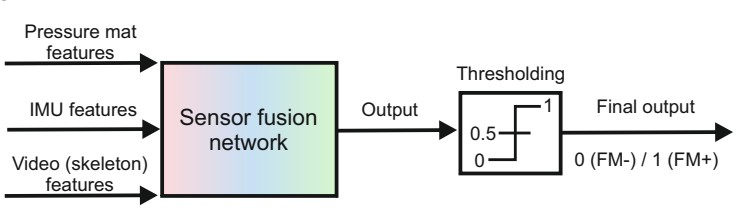

We denote accelerometer and gyrometer values as $x^a, y^a, z^a$, and $x^g, y^g, z^g$, respectively (see Fig. 2c). We applied moving average filter for each sequence with a sliding window size of 5 frames, and then centered each time series by subtracting mean value of the corresponding time series.

The acceleration values and angular velocity values are of different scale, thus, we normalized acceleration values and angular velocity values separately using z-score normalization. For this, we calculated mean and standard deviation values across all acceleration (or angular velocity) time series in training sets and then used these values to normalize data in the training and test sets.

After pre-processing, and concatenation of acceleration and angular velocity values, this led to the feature matrix of size $300 \times 36$:

$$\begin{bmatrix} x_1^{a_1} & y_1^{a_1} & z_1^{a_1} & x_1^{g_1} & y_1^{g_1} & z_1^{g_1} & \cdots & x_1^{a_6} & y_1^{a_6} & z_1^{a_6} & x_1^{g_6} & y_1^{g_6} & z_1^{g_6} \\ x_2^{a_1} & y_2^{a_1} & z_2^{a_1} & x_2^{g_1} & y_2^{g_1} & z_2^{g_1} & \cdots & x_2^{a_6} & y_2^{a_6} & z_2^{a_6} & x_2^{g_6} & y_2^{g_6} & z_2^{g_6} \\ \vdots & \vdots & \vdots & \vdots & \vdots & \vdots & \ddots & \vdots & \vdots & \vdots & \vdots & \vdots & \vdots \\ x_n^{a_1} & y_n^{a_1} & z_n^{a_1} & x_n^{g_1} & y_n^{g_1} & z_n^{g_1} & \cdots & x_n^{a_6} & y_n^{a_6} & z_n^{a_6} & x_n^{g_6} & y_n^{g_6} & z_n^{g_6} \end{bmatrix},$$

(9)

with $n = 300$ frames.

**Classification models**

In this study, we were dealing with a binary classification task where we classified fidgety movements (FM+) vs. non-fidgety movements (FM-).

**Neural network architectures.** For the comparison of classification performance using different sensor modalities, we used convolutional neural network (CNN) architectures with three convolutional layers (Conv) and one fully connected (FC) layer (see Fig. 3a). Each convolutional (Conv) and fully connected (FC) layer was followed by a batch normalization layer and a drop-out layer (20%). ReLU activation functions were used in the Conv and FC layers, whereas in the output layer a linear activation function was used.

To train neural classifiers, we used binary cross-entropy with *logit* transfer function as a loss function (therefore linear activation function in the output layer) and the Adam optimizer with the following parameters: batch size 4, learning rate = 0.001, $\beta_1 = 0.9$, $\beta_2 = 0.999$, and $\epsilon = 1e$-07. To avoid model overfitting, we used a validation stop with the validation split 1/8 and patience of 10 epochs. The network architectures were implemented using TensorFlow[56] and Keras API[57].

**Hyperparameter tuning.** To tune hyperparameters of the network architectures we used a separate dataset where snippets from 9 infants were used (data from these infants was not used in the 9-fold cross-validation procedure). The data was subdivided into training and validation sets. The number of snippets used for hyperparamter tuning is given in Table 1.

We tuned hyperparameters of the network architectures for each sensor modality and the combination of all sensor modalities using a two-stage procedure. First, we tuned the number of convolutional (Conv) and

**Table 1 | Data split for the hyperparameter tuning and 9-fold cross-validation**

| Hyperparameter tuning (9 infants) | | | | | | | |
|---|---|---|---|---|---|---|---|
| | FM- | FM+ | Total | # snippets per infant: | | | |
| | | | | Mean (SD), [Min Max] | | | |
| | 148 | 189 | 337 | 37 (31), [7 106] | | | |

| 9-fold cross-validation (36 infants) | | | | | | | | |
|---|---|---|---|---|---|---|---|---|
| | Training set (32 infants) | | | | Test set (4 infants) | | | |
| Fold # | FM- | FM+ | Total | # snippets per infant: | FM- | FM+ | Total | # snippets per infant: |
| | | | | Mean (SD), [Min Max] | FM- | FM+ | Total | Mean (SD), [Min Max] |
| 1 | 519 | 671 | 1190 | 37 (24), [4 87] | 73 | 83 | 156 | 39 (21), [10 61] |
| 2 | 537 | 666 | 1203 | 38 (22), [4 85] | 55 | 88 | 143 | 36 (36), [7 87] |
| 3 | 523 | 673 | 1196 | 37 (23), [4 87] | 69 | 81 | 150 | 38 (24), [16 68] |
| 4 | 524 | 665 | 1189 | 37 (24), [4 87] | 68 | 89 | 157 | 39 (17), [18 59] |
| 5 | 533 | 674 | 1207 | 38 (23), [7 87] | 59 | 80 | 139 | 35 (26), [4 65] |
| 6 | 521 | 668 | 1189 | 37 (23), [4 87] | 71 | 86 | 157 | 39 (31), [20 85] |
| 7 | 526 | 671 | 1197 | 37 (22), [4 87] | 66 | 83 | 149 | 37 (36), [7 79] |
| 8 | 530 | 672 | 1202 | 38 (24), [4 87] | 62 | 82 | 144 | 36 (23), [17 68] |
| 9 | 523 | 672 | 1195 | 37 (24), [4 87] | 69 | 82 | 151 | 38 (11), [24 50] |

Whole dataset consists of 1683 snippets (740 FM- and 943 FM+) obtained from 45 infants. 337 snippets (obtained from 9 infants) were used for the hyperparameter tuning and 1346 snippets (obtained from 36 infants) were used for the cross-validation.

fully connected (FC) layers, the number and size of kernels in the Conv layers, and the number of units in the FC layers using the Bayesian optimization. In the first stage, we explored architectures with 1, 2 or 3 Conv, and 1 or 2 FC layers. Other hyperparameters for the Bayesian optimization procedure are given in Supplementary Table 1. We ran Bayesian optimization for 1000 steps, and repeated this optimization procedure five times. We trained our networks using validation stop with a validation split of 1/5 and patience of 10 epochs. The other training parameters were the same as given in section "Neural network architectures". After the optimization procedure, we selected the 10 different best models with lowest loss scores on the validation set, and analyzed hyperparameters of these best models. In most of the cases, architectures with three Conv layers and one FC layer were obtained.

In the second stage, we performed fine-tuning using grid search, where we fixed the number of Conv and FC layers (three and one, respectively) based on the results of the Bayesian optimization procedure, and only tuned the number and size of the kernels in the Conv layers and the number of units in the FC layer within the reduced hyperparameter space (see Supplementary Table 1). We repeated the grid search three times, where in each repetition we explored 4800 different hyperparameter sets. Finally, we selected 10 different best models (see Supplementary Tables 2 and 3) with the lowest loss scores on the validation set. These models then were used for the 9-fold cross-validation procedure.

The hyperparameter tuning was implemented and performed using KerasTuner[58].

**Sensor fusion**

We tested and compared two sensor fusion approaches. In the first approach (late fusion, see Fig. 3b), we combined outputs of the convolutional neural networks as shown in Fig. 3a, which were trained on single sensor modalities: MAT network trained using pressure features extracted from the pressure mat data, IMU network trained using IMU signals, and VID network trained using skeleton key points extracted from the video data. Each network outputs a value between 0 and 1, which corresponds to the probability $p$ for the class FM+ and $1 - p$ for the class FM-. To obtain the final decision, we computed the average probability of two (combination of two sensor modalities) or three (combination of three sensor modalities) networks, and then applied a threshold of 0.5 to obtain class label 0 or 1 for the FM- or FM+ class, respectively.

In the second approach (early fusion, see Fig. 3c), we trained one convolutional neural network (see box Sensor fusion network) where we used concatenated features from all sensor modalities as one feature matrix. To ensure temporal compatibility (i.e., number of frames per snippet) of different sensor modalities, we down-sampled time series of the pressure mat and IMU features to 250 frames to make it consistent with the sampling rate of the video data. After re-sampling and concatenating three feature matrices, we obtained the final feature matrix of size $250 \times (6+36+60) = 250 \times 102$. The output of the Sensor fusion network is the probability $p$ for the class FM+ and $1 - p$ for the class FM-. As in case of the late fusion approach (see Fig. 3b), to obtain class label 0 or 1 for the FM- or FM+ class, respectively, we applied a threshold of 0.5. To tune and train the Sensor fusion network, we used the same procedures for hyperparameter tuning and training as explained in sections "Neural network architectures" and "Hyperparameter tuning".

**Statistics and reproducibility**

For the evaluation and comparison of the classification models, we used a 9-fold cross-validation procedure where in total data from 36 infants were used. For this, we divided the dataset into 9 subsets where each subset contained snippets from 4 different infants. For each fold, one subset was used as the test set, and the remaining eight subsets (snippets from 32 infants) were used to train the network architectures. The number of snippets in the training and test sets for each fold is given in Table 1.

In total we trained and tested 10 best models for each single sensor modality (see Supplementary Table 2) and the combination of all sensor modalities (see Supplementary Table 3). For training, we split the training set into training (7/8 = 87.5%) of the training data) and validation (1/8 = 12.5%) of the training data) subsets. For each fold we trained each model 20 times (with random initial conditions) and then selected the model with the lowest loss score on the validation set, which was then evaluated on the test set.

For the comparison of the classification performances, we used three common classification performance measures: *sensitivity* (true positive rate – TPR), *specificity* (true negative rate – TNR) and *balanced accuracy* – BA:

$$TPR = \frac{TP}{TP + FN}, \quad TNR = \frac{TN}{TN + FP}, \quad BA = \frac{TPR + TNR}{2}, \quad (10)$$

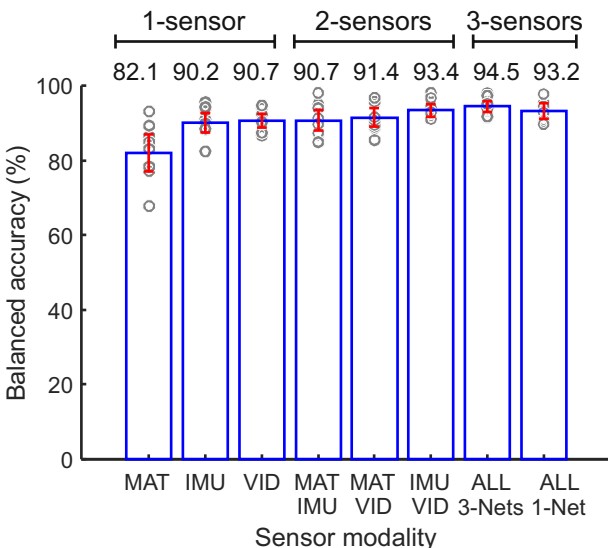

**Fig. 4 | Comparison of classification performances of the best models of different sensor modalities on the test sets.** Average balanced classification accuracy obtained from 9-fold cross-validation is shown for each case ($n = 9$). Error bars denote confidence intervals of mean (CI 95%). Gray circles denote classification accuracies for each test set.

where $TP$ is the number of true positives, $TN$ the number of true negatives, $FP$ the number of false positives, and $FN$ the number of false negatives.

To compare classification accuracies of the classification models, we calculated average classification performance measures (sensitivity [$TPR$], specificity [$TNR$], and balanced accuracy [$BA$]) across nine test sets, confidence intervals of mean (CI 95%), and $p$ values for the comparison of medians using the Wilcoxon two-sided signed-rank test. Statistical significance was set at $p < 0.05$.

Data and code are publicly available at Zenodo[59].

### Reporting summary
Further information on research design is available in the Nature Portfolio Reporting Summary linked to this article.

## Results
### Classification performance using single sensor modalities
Comparison of the classification performance when using only one sensor modality is shown in Fig. 4 and Tables 2 and 3. Note that here we compare results of the best models (out of 10). The results for all models are shown in Supplementary Table 4. The classification performance using pressure mat features (MAT, 82.1%) was lower than IMU (90.2%; approached significance, $p = 0.055$), and significantly lower than skeleton (video-based) features (VID, 90.7%; $p < 0.01$; Table 3). The performances of IMU and VID were not significantly different from each other.

The span of the classification accuracies using pressure mat sensor across nine test sets was large (CI 95% = [77.1% 87.0%]), which may imply that the current mat only worked well for fidgety movement classification for some infants, but not for the others. Classification accuracies using IMU and video sensors were stable across all test sets, with smaller confidence intervals, [87.6% 92.8%] and [88.9% 92.4%], respectively.

### Classification performance using sensor fusion
Regarding two-sensor fusions (MAT+IMU, MAT+VID, IMU+VID), the classification accuracies were generally higher than those of the single modality (Table 2), although the differences were rarely statistically significant (Table 3). In particular, when fusing VID with IMU, the performance was significantly superior to that of MAT or IMU alone, but not compared to VID alone. When combining MAT with either IMU or VID,

the performances were indeed significantly better than that of MAT alone, but not than IMU or VID alone. That is, adding the available sensor-mat to one of the other two sensors did not bring significant improvement than using the IMUs or the camera alone for the present task.

The performance of the three-fold fusion (the combination of three networks [ALL 3-Nets] considered) was superior to any of the single sensor alone (see Fig. 4 and Tables 2 and 3). While the three-sensor fusion approach was also significantly better than the two-sensor fusions models where MAT was involved (i.e., MAT+IMU or MAT+VID), it was not superior to the combination of VID+IMU. Note, although VID+IMU or VID+MAT was not significantly better than VID alone, combining all three modalities was superior to VID alone.

We also tested whether using features of all sensor modalities as input for one network (early fusion, ALL 1-Net, see Supplementary Table 5) would lead to a better/worse classification accuracy as compared to the combination of three networks trained on the single sensor modalities (late fusion, ALL 3-Nets). While the average classification accuracy of the 3-Nets model (94.5%) was higher than that of the 1-Net model (93.2%), the difference was not statistically significant ($p = 0.164$).

## Discussion
Following our experience in the use of mono-sensor approaches in the study of infant motor functions, we now propose a multimodal approach for movement recognition and classification. Using a fully synchronized multi-sensor dataset, in the present study we empirically tested and compared the utilities of three sensor modalities (pressure sensor, MAT; inertial sensors, IMU; and visual sensor, VID) and their various combinations (two- and three-sensor fusion approaches) for the same task. The task was to track and classify age-specific general movement patterns and differentiate between the absence or presence of a specific motor pattern during the first months of life, non-fidgety (FM-) vs. fidgety (FM+) movements[8,9,44–46,60].

To our first question, the performance of the single sensor modalities was different, with MAT (82.1%) being inferior to IMU (90.2%, $p = 0.055$) and VID (90.7%, $p < 0.01$). The performances of IMU and VID were, however, comparable, and also comparable to the average accuracy of divergent automated approaches for GMA studied to date[7,26,28,29,61]. Nevertheless, these results were generated and validated on silo-data sets which do not allow for direct comparison of the findings.

The second methodological research question was whether sensor fusion improves classification accuracy. With the sensors available at the time of study conduct and used specifically for this task, the performance of the three-sensor fusion (94.5%) was significantly more accurate than any of the single modalities. It was also significantly superior than the two-sensor fusions of MAT+IMU (90.7%) or MAT+VID (91.4%). The average accuracy of the three-sensor fusion approach was again higher than the two-fold fusion of VID+IMU (93.4%), although the difference was not significant ($p = 0.22$). The two-sensor fusion approaches with MAT (MAT+VID and MAT+IMU) were both significantly better than the performance of MAT alone; and the fusion with VID+IMU outperformed both MAT or IMU alone. With both two- and three-sensor fusions, the performances were superior than those of their single components, although some values were not statistically significant (Tables 2 and 3).

In our previous work, we extensively discussed possible reasons why MAT delivered moderate classification accuracy in our experiments, and the pros and cons of using divergent sensor modalities for infant motion tracking and classification[21]. With rapid technology advancement, the performance of each single modality (e.g., MAT) is likely to improve, which may further increase the accuracy of combined modalities (sensor fusion). This, however, needs to be tested in the future. For any specific task and with any single modality, the classification accuracy will eventually reach its limit (either ceiling or bottle neck). It is critical to examine, whether an approach with sensor fusion might be unnecessary (by perfect or ceiling accuracy of a single modality) or whether it is beneficial (by bottle neck of a single modality). If different dimensions of input (from diverse modalities) for the same phenomenon (e.g., motion) are augmented, the representation of the

**Table 2 | Comparison of classification performances of the best models of different sensor modalities on the test sets**

| Model | Sens. (%) [CI] | Spec. (%) [CI] | BA (%) [CI] |
|---|---|---|---|
| Single sensor modalities (best models) | | | |
| MAT | 86.17 [82.78 89.55] | 77.95 [69.68 86.22] | 82.06 [77.11 87.00] |
| IMU | 92.91 [90.28 95.55] | 87.52 [83.47 91.57] | 90.22 [87.61 92.82] |
| VID | 91.67 [89.80 93.55] | 89.65 [85.89 93.41] | 90.66 [88.91 92.41] |
| Sensor fusion (best models) | | | |
| MAT+IMU | 94.12 [92.56 95.68] | 87.36 [82.67 92.05] | 90.74 [87.95 93.53] |
| MAT+VID | 92.58 [89.53 95.62] | 90.27 [86.50 94.04] | 91.42 [88.93 93.91] |
| IMU+VID | 94.43 [92.53 96.34] | 92.29 [89.59 94.99] | 93.36 [91.75 94.97] |
| ALL 3-Nets | 96.16 [95.03 97.29] | 92.85 [90.23 95.46] | 94.50 [93.05 95.95] |
| ALL 1-Net | 92.87 [88.45 97.29] | 93.60 [91.01 96.19] | 93.24 [91.15 95.32] |
| Sens. – Sensitivity, Spec. – Specificity, BA – Balanced accuracy. | | | |

Average classification measures together with confidence intervals of mean (CI 95%) obtained from 9-fold cross-validation ($n = 9$).

**Table 3 | $p$-values for the pairwise comparisons of the classification accuracies (balanced accuracy) of different models**

| | IMU | VID | MAT+IMU | MAT+VID | IMU+VID | ALL 3-Nets | ALL 1-Net |
|---|---|---|---|---|---|---|---|
| MAT | 0.0547 | 0.0039** | 0.0078** | 0.0039** | 0.0078** | 0.0039** | 0.0078** |
| IMU | - | 0.9102 | 0.9102 | 1.0000 | 0.0078** | 0.0039** | 0.0078** |
| VID | - | - | 1.0000 | 0.3594 | 0.0547 | 0.0117* | 0.3008 |
| MAT+IMU | - | - | - | 0.4258 | 0.0742 | 0.0117* | 0.0391* |
| MAT+VID | - | - | - | - | 0.1289 | 0.0195* | 0.3008 |
| IMU+VID | - | - | - | - | - | 0.2188 | 1.0000 |
| ALL 3-Nets | - | - | - | - | - | - | 0.1641 |
| Asterisk symbols denote significant difference at the confidence level $a = 0.05$: *$p < 0.05$, **$p < 0.01$. | | | | | | | |

Wilcoxon two-sided signed-rank test ($n = 9$).

ground truth (e.g., distinguishable patterns) may be improved, hence increasing the overall classification accuracy, as shown in our current study. If, however, different modalities only provide redundant and highly correlated information in similar dimensions, then the combined approach may not outperform its single components.

All available automated GMA approaches solve only a fraction of the tasks that are involved in a human GMA[21]. This is legitimate and fundamental for the beginning of tool development, especially for proof-of-concept and method development endeavors. However, an AI tool, if intended to be used for real-time clinical practice not only for detecting cerebral palsy, needs to go beyond providing dichotomous labels for a simplified task (e.g., FM+ or FM-), and with data from a limited sample. When more complicated tasks are demanded (e.g., divergent age-specific normal and inconspicuous movements vs. divergent age-inadequate or aberrant movements, each pointing to different clinical outcomes), it is important to evaluate again if a single modality method is sufficient, or, whether approaches with sensor fusion would be needed and generate rewarding results.

The third important question is whether sensor fusion approaches with non-intrusive modalities yield sufficient classification performance. Recall that GMA is not only valued for its efficiency and accuracy, but even more so for its convenience and non-intrusiveness to infants and their families. This is the reason why it has been widely accepted by families of divergent cultural backgrounds[62], and embedded in clinical routines. Note that a high number of infants in need of a GMA diagnosis are in poor or delicate health conditions. Attaching sensors to the infants' body is challenging, and, as known, could interfere with their behavioral states and hence their motor output (handbook[49]). In a standard GMA, the assessor does not need to touch, manipulate or neurologically assess, but only observe the infant for a few minutes. If an automated solution aims to scale-up GMA in medical practice, it should maintain the non-intrusiveness and user-friendliness of the original method[8]. In this study, the non-intrusive sensor fusion (MAT+VID) resulted in satisfying and significantly better accuracy (91.42%) than MAT alone (82.1%), and higher (yet not significant) accuracy than VID (90.7%). In other words, the fusion with non-intrusive sensors delivered promising results. We cannot tell at this point if the performance of MAT could be further improved (e.g., with refined technology customized for infant motion tracking), and whether the combined performance of MAT+VID would then change and surpass that of VID alone.

Notably, the pressure sensing mat is fully non-intrusive and effortless to install. It is also anonymous by nature, hence particularly suitable for acquiring and sharing multi-centered large-scaled clinical data in a short time, which may be the key for developing and advancing any data-driven AI approach[21]. That said, improving pressure sensing technology for infant motion tracking might be easier and faster than improving the technology of other sensing modalities. A single camera, as used in this study, is also non-intrusive and without complicated setups, and is a widely used and researched modality for automated GMA solutions[1–3]. However, compared to pressure sensors, cameras use implies confidentiality issues, which may hinder large scale data sharing. This must be circumvented (see for an example[45]) and is associated with additional costs and efforts. At present, applying intrusive wearable devices such as IMUs on the infant's body is still cumbersome and time-consuming. Still, future IMUs might be developed for effortless and user-friendly applications, which will make this sensing method more attractive. In our study, IMUs yielded a performance comparable to RGB cameras.

For developing the sensor-fusion method, we used a dataset of 1683 snippets from 45 infants, fully labeled by two human GMA assessors.

The performances of the single sensor modalities and their combinations may be improved if larger datasets are used in future undertakings. Extended datasets would also allow the use of more advanced network architectures such as graph convolutional neural networks[26], spatio-temporal attention models[28], or spatial-temporal transformer models[7]. In addition, the utility of 3D skeletons instead of 2D skeletons[7,30], or RGB-D sensors[27,30], and the utility of ensemble classifiers[26,61] may improve the classification performance of the fusion approach with non-intrusive sensors even further.

In our study, we compared two sensor fusion approaches: a combination of three networks trained on single sensor modalities vs. one network trained on all sensors (early vs. late sensor fusion, e.g., see[63]). While on average classification accuracy of the late sensor fusion was slightly higher than that of the early fusion (94.5% and 93.2%, respectively), this difference in the classification performance was not statistically significant. However, we would argue that using the late sensor fusion approach (i.e., the combination of multiple networks) would be more advantageous. The first reason is that a modular approach (three separate networks) makes it easier to retrain or replace and retrain classifiers for the specific sensor modality. The feature space of single sensor modalities is much smaller as compared to the feature space of a combination of all the sensors. Larger feature spaces usually require larger networks, which leads to longer training time and may cause convergence issues. The other reason is that the single network would break down if during a multi-sensory recording one of the sensors failed (i.e., missing data from one sensor), since inputs from all sensors are required for making predictions.

Last but not least, the sensor fusion approach currently needs complex setups and data analysis procedures as compared to single components, therefore it is associated with higher costs, greater efforts, and more technical requirements. If, however, the sensor fusion approach, especially the non-intrusive ones, could decisively improve the performance of automated solutions of GMA compared to that of the single modalities, the efforts would ultimately pay off. The sensor fusion approach is yet another attempt among many others developed over the past decades, aimed at generating automated GMA modalities with high accuracy and superior user experiences.

In conclusion, the multiple sensor fusion approach is a promising attempt for automating the classification of infant motor patterns. Complex evolving neurofunctions require equally complex (and very likely multi-sensor) assessments enabling a deeper understanding of infant development. We consider the multiple sensor fusion approach an essential methodological challenge towards realizing AI-based deep phenotyping and classification of infant movements. GMA studies and infant research over past decades suggest that age-specific endogenous movements are associated with different neurodevelopmental outcomes across distinct domains such as motor, social-communicative and cognitive functions[9,44,64,65]. The multiple sensor fusion approach will add a broader perspective to the prevailing single modality endeavors focusing on the recognition of a selected number of movement patterns[2,7,21,26,38,39,66]. This work catalyzes further innovation, driving empirical studies to enable AI based solutions to effectively predict developmental trajectories associated with diverse outcomes.

## Data availability
The data used for the classification experiments is publicly available at Zenodo: https://doi.org/10.5281/zenodo.14046527[59]. The source data underlying results presented in Fig. 4 is available in the Supplementary Data 1. The source data underlying results presented in Supplementary Fig. 1 and Supplementary Fig. 2 are available in the Supplementary Data 2. All other data are available from the corresponding author on reasonable request.

## Code availability
The code is publicly available at Zenodo: https://doi.org/10.5281/zenodo.14046527[59]. Python scripts were tested on Linux OS with TensorFlow 2.10.1,

Keras 2.10.0, Python 3.7.6, and Numpy 1.21.6. Matlab scripts were tested on Windows OS with MATLAB R2019b.

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

## Acknowledgements

We would like to especially dedicate this work to Christa Einspieler, having developed this idea together, who is currently undergoing challenging times. The authors thank all families for their participation in this close-meshed prospective study. The initial idea of this study was conceptualized together with the "father of GMA", Heinz Prechtl, and one of the leading experts in spontaneous infant movement assessment, Christa Einspieler. The transition from a Gestalt to an AI-based GMA would not have been possible without their guidance and skepticism on our initial ideas which was essential to develop the suggested methodology. Special thanks to the team members of the Marschik-Labs in Graz and Göttingen (iDN–interdisciplinary Developmental Neuroscience and SEE–Systemic Ethology and Developmental Science), who were involved in recruitment, accompanying families and their babies, data acquisition, data curation and pre-processing, and technical support: Gunter Vogrinec, Magdalena Krieber-Tomantschger, Iris Tomantschger, Laura Langmann, Claudia Zitta, Dr. Robert Peharz, Dr. Florian Pokorny, all student assistants, and Dr. Simon Reich. Special thanks also to our interdisciplinary international network of collaborators for discussing this study with us and for refining our views and ideas. We are/were supported by Bio-TechMed Graz and the Deutsche Forschungsgemeinschaft (DFG – stand-alone grant 456967546, SFB1528 – projects B01, C03, Z02), the Bill and Melinda Gates Foundation through a Grand Challenges Explorations Award (OPP 1128871), the Volkswagen Foundation (project IDENTIFIED), the Leibniz ScienceCampus, the BMBF CP-Diadem (Germany), and the Austrian Science Fund (KLI811) for data acquisition, preparation, and analyses. We acknowledge support by the Open Access Publication Funds/transformative agreements of the Göttingen University.

## Author contributions

Conceptualization, T.K., D.Z., and P.B.M.; Methodology, T.K., D.Z., F.W. and P.B.M; Data curation, T.K. and S.F.; Implementation and software, T.K.; Formal analysis, T.K. and D.Z.; Visualization, T.K.; Writing: Original draft, T.K., D.Z., F.W., and P.B.M.; Writing: Revision and editing, T.K., D.Z., L.P., S.B., L.J., M.K., M.Z., K.N., F.W. and P.B.M.; Supervision, F.W. and P.B.M.

## Funding

## Competing interests

The authors declare no competing interests.

## Additional information

[1]Child and Adolescent Psychiatry and Psychotherapy, University Medical Center Göttingen, Leibniz ScienceCampus Primate Cognition and German Center for Child and Adolescent Health (DZKJ), Göttingen, Germany. [2]Department of Child and Adolescent Psychiatry, University Hospital Heidelberg, Heidelberg University, Heidelberg, Germany. [3]Department for Computational Neuroscience, Third Institute of Physics - Biophysics, Georg-August-University of Göttingen, Göttingen, Germany. [4]iDN – interdisciplinary Developmental Neuroscience, Division of Phoniatrics, Medical University of Graz, Graz, Austria. [5]Center of Neurodevelopmental Disorders (KIND), Department of Women's and Children's Health, Center for Psychiatry Research, Karolinska Institutet & Region Stockholm, Stockholm, Sweden. [6]Child and Adolescent Psychiatry, Stockholm Health Care Services, Stockholm, Sweden. [7]Curtin Autism Research Group, Curtin School of Allied Health, Curtin University, Perth, Australia. [8]Department of Medical Engineering, Technical University Berlin, Berlin, Germany. [9]German Center for Neurodegenerative Diseases (DZNE), Göttingen, Germany. [10]Department for NMR-based Structural Biology, Max Planck Institute for Multidisciplinary Sciences, Göttingen, Germany. [11]Department of Pediatrics, David Geffen UCLA School of Medicine, Los Angeles, CA, USA. [12]These authors contributed equally: Tomas Kulvicius, Dajie Zhang. [13]These authors jointly supervised this work: Florentin Wörgötter, Peter B Marschik. ✉e-mail: tomas.kulvicius@med.uni-goettingen.de; peter.marschik@med.uni-heidelberg.de

