## [Transparent Peer Review file · Communications Medicine]

Deep learning empowered sensor fusion boosts infant movement classification

Corresponding Author: Dr Tomas Kulvicius

Version 0:

Reviewer comments:

Reviewer #1

(Remarks to the Author)

In this study, the authors developed a sensor fusion approach and performed a deep learning comparison of three different sensor modalities for classification of infant spontaneous movements during the first months of life. Their aim is to increase accuracy of automated General Movement Assessment approaches using three different modality methods for classification of fidgety movements (FMs). The authors use convolutional neural networks (CNN) on spontaneous movement between 28- and 112-days post-term age in 51 infants comparing different multi-sensory movement recording setup with different combinations of video cameras, pressure sensing devices and inertial measurement units (IMUs) attached to the infant body using a body-suit.

The authors ability to evaluate and discuss the different sensor modalities accuracy for classification of FMs is novel and would be very helpful in the design of further clinical studies, overall aiming at automated GMA and early prediction of cerebral palsy (CP), which could easily be implemented in clinically settings. The study is convincing and will influence thinking by people in the field of early detection of cerebral palsy and automatic infant motion capture. The manuscript is well written, and the methodology is sound. I recommend publication.

Please find the following comments:

Introduction:

1) The diagnosis of CP is made clinically using many different assessment tools (not only MRI that is mentioned by the authors). The GMA is one of them. A broader description of the clinical context where automated GMA potentially will have a future – short – is missing in the beginning of the introduction.

2) The authors describe in the introduction that classification performance of other automated GMA methods using single modality methods have an accuracy level about 90%. They claim that this is considerably inferior to that of well-trained human assessors (GMA experts?). Please moderate the statement “considerably inferior”, since several studies using the GMA in clinical settings has shown lower accuracy levels than 90% and in particular with respect to positive and negative predictive values which is the most important predictive values using an assessment tool in clinical settings.

Methods:

3) Second paragraph, 2.2.3 Movement annotations: Description of labeling snippets as “FM+”, FM-“. Please provide details about were FM+/- and FM++ belongs into these categories and provide the same in legend Figure 1.

4) First paragraph, 2.2.1 Participants: Description of infant inclusion criteria. Please use the term Typically Developing (TD) infants when listing up details.

5) First and second paragraph, 2.2.2 Multi-modal movement recordings: In first paragraph the authors use “... days post term age”, in second paragraph “...weeks post term age”. Please be consistent.

6) Last sentences, 2.2.2 Multi-modal movement recordings: The authors describe a customized body-suit with designated pockets for IMU sensors. Customized for each individual infant? Was the body-suit customized in such a way that the IMU sensors did not move in relation of the infant body? Please clarify.

7) Second paragraph, 2.2.3 Movement annotations: In the brackets, the authors define the labelling of “not assessable”. Please provide details about how “over-exited” is defined.

Discussion:

8) As far as I can understand, the performance of the different sensor fusion modalities range between 82.1% and 90.7% using a single modality and between 90.7% and 93.4% using several modalities. Please provide clearer somewhere in the discussion – short - what clinical implications the findings might have, if any?, and how this relates to the clinical context that is described in the introduction (related to comment 1)). Maybe put it in somewhere in relation to the descriptions in last part of paragraph 4 in the Discussion?

9) Related to comment 8) about implications for clinical practice. In paragraph 5 the authors claim that an AI tool used in real-time clinical practice, needs to go beyond providing dichotomous labels (FM+ or FM-). Why? Today, expert-based GMA is used in such a way for prediction of CP. Why is this not the case for a possible AI tool? Please elaborate or clarify a bit if possible.

Reviewer #2

(Remarks to the Author)

Introduction: The authors effectively identify the 'gold standard' of General Movements Assessment (GMA) and highlight the superiority of assessments conducted by trained human assessors, as well as the limitations that currently impede the adoption of AI-driven GMA. The introduction of a sensor fusion approach using Inertial Measurement Units (IMUs) is innovative and successfully engages the reader with the work. The writing is accessible to a broad audience, well-referenced, and the research objectives are clearly articulated. However, the introduction does not sufficiently address the rationale for developing an AI solution when trained human analysis is considered superior. I recommend that the authors include a discussion on the limitations of relying solely on trained human assessors and assessments. This addition would clarify the necessity of an AI solution in addressing broader population health challenges. In the current state, the argument for AI's role, particularly within the infant provider community, lacks the compelling urgency required to challenge established care principles that have persisted for decades.

Methods: The methods section appropriately considers the time horizon of fidgety movement emergence and the optimal detection period. However, there is no information on the qualifications and experience of the two assessors, which is important for understanding the reliability of the human assessment component, especially since the IRR deteriorated upon a second analysis in some cases. Additionally, the use of a handheld HD camcorder raises questions, as a smartphone could achieve similar resolution while offering greater applicability in underserved areas. Providing justification of this selection is recommended. In section 2.4, "Neural Network Architectures," there is a typo in the first statement ("were we classified"), which should be corrected. The overall methodology is clear, well-described, and replicable. The accompanying graphics and tables contribute meaningfully to the understanding of the research.

Results: The results are clearly presented, well-supported by the data, and the analysis is robust and thorough. I have no further suggestions for this section.

Discussion: The discussion provides insightful and comprehensive analysis of the findings and proposes directions for future work. I suggest minimally expanding on the adoption issues impacting each modality (or modality combination). This would not only inform future research and technical innovation but also emphasize accessibility for a broader range of systems and populations that could benefit which is primary to the aims and scope of this publication.

Overall Recommendation: I consider this manuscript to be a significant contribution to the literature on AI, GMA, and infant movement classification. I support its publication with minor revisions as outlined above. GMA is a critical component in clinical pathways, especially in neonatal intensive care units (NICUs), yet access to and training of expert providers remain barriers to widespread adoption. AI presents a promising and innovative strategy to overcome these challenges.

Reviewed by: Teresa Fair-Field, OTD, OTR/L (GMA-certified occupational therapist and infant movement classification researcher specializing in machine learning techniques)

Reviewer #3

(Remarks to the Author)

The study titled "Deep learning empowered sensor fusion boosts infant movement classification" explores the application of AI - and specifically deep learning - in enhancing diagnostic processes using the Prechtl general movement assessment (GMA) to detect neurological impairments in infants. It specifically investigates a sensor fusion approach using pressure, inertial, and vision sensors to assess infant movements. The study compares various sensor combinations and fusion techniques (late and early fusion) to determine if a multi-sensor system can outperform single-sensor modalities. The findings reveal that the three-sensor fusion achieves a significantly higher classification accuracy (94.5%).

The paper is well written and is relevant from a scientific and clinical point of view. I have some concerns that I report below:

- It is not clear from the abstract what classification method has been implemented. Also, the first sentence in the abstract seems disconnected and uninformative from the rest of the text so I suggest removing it.
- I suggest concluding the introduction with a bulleted list of the innovative contributions of this work.
- To improve readability the state of the art should occupy a single Section in which for each approach mentioned the

limitations are analysed. Literature in the field of decision support systems based on deep learning should be up-to-date (i.e., 2021-2022-2023-2024).

- Authors are encouraged to share codes and data for the reproducibility of the experiments from other researchers.
- What criteria will determine the release of the dataset? Will it be accessible to all researchers? Will the entire dataset be made available? I think that sharing this data with research groups in the field is crucial, as it represents a significant contribution of this work, especially since the proposed deep learning methodology itself does not introduce any innovations.
- Given the focus of the work, it would be important to have (1) a paragraph that well explains how multi-source data fusion occurs (the sensor fusion paragraph seems poorly informative), (2) how the input is secured to the architecture. All the blocks in Figure 3 should be explained in the text.
- How was the data used synchronized?
- It would be important to report a summary table of the training/testing and validation sets making it clear how much data from individual patients are in each of the sets (it is crucial to have a patient split for cross-validation purposes. Has this been implemented?).
- I would avoid using the accuracy metric (Figure 4). Rather, I would plot and discuss the confusion matrices.

Version 1:

Reviewer comments:

Reviewer #1

(Remarks to the Author)

I have reviewed all changes and amendments made by the authors according to my comments and are satisfied and happy with all reponses. The manuscript now fulfils the standard needed for publication.

Reviewer #2

(Remarks to the Author)

concerns have been satisfied.

Reviewer #3

(Remarks to the Author)

I thank the Authors for addressing my concerns, I have no further questions.

List of Changes

Dear Editor, dear Reviewers,

we sincerely thank you for the constructive comments and valuable suggestions which have helped us to significantly improve our manuscript. We have thoroughly and carefully addressed all issues raised (please see point-by-point replies below) and hope that our manuscript meets the reviewers' expectations and the high standards of your journal.

We marked our **replies** in **blue**. **Changes** in the manuscript are highlighted in **red**.

Thank you very much, indeed.

Yours sincerely,

Tomas Kulvicius on behalf of all authors

Referee expertise

Referee #1: fidgety movements, general movement assessment, prognosis, early detection, high-risk infant

Referee #2: infant assessment, machine learning, motor development, validity, population-level smartphone app, infant motor screening, parent-led video

Referee #3: deep learning, preterm infants, smart walker, movement monitoring

Reviewers' comments

Reviewer #1 (Remarks to the Author)

In this study, the authors developed a sensor fusion approach and performed a deep learning comparison of three different sensor modalities for classification of infant spontaneous movements during the first months of life. Their aim is to increase accuracy of automated General Movement Assessment approaches using three different modality methods for classification of fidgety movements (FMs). The authors use convolutional neural networks (CNN) on spontaneous movement between 28- and 112-days post-term age in 51 infants comparing different multi-sensory movement recording setup with different combinations of video cameras, pressure sensing devices and inertial measurement units (IMUs) attached to the infant body using a body-suit.

The authors ability to evaluate and discuss the different sensor modalities accuracy for classification of FMs is novel and would be very helpful in the design of further clinical studies, overall aiming at automated GMA and early prediction of cerebral palsy (CP), which could easily be implemented in clinically settings. The study is convincing and will influence thinking by people in the field of early detection of cerebral palsy and automatic infant motion capture. The manuscript is well written, and the methodology is sound. I recommend publication.

Reply: Thank you very much for your positive evaluation and feedback. We are excited about your review and grateful for the constructive comments. We have addressed these issues point-by-point below.

Please find the following comments:

Introduction:

1) The diagnosis of CP is made clinically using many different assessment tools (not only MRI that is mentioned by the authors). The GMA is one of them. A broader description of the clinical context where automated GMA potentially will have a future – short – is missing in the beginning of the introduction.

Reply: Thank you for pointing this out. We have added a few lines to the Introduction, first paragraph, to address this.

2) The authors describe in the introduction that classification performance of other automated GMA methods using single modality methods have an accuracy level about 90%. They claim that this is considerably inferior to that of well-trained human assessors (GMA experts?). Please moderate the statement “considerably inferior”, since several studies using the GMA in clinical settings has shown lower accuracy levels than 90% and in particular with respect to positive and negative predictive values which is the most important predictive values using an assessment tool in clinical settings.

Reply: We agree and have deleted the first part of this sentence and now only focus on the non-comparability of automated approaches.

Methods:

3) Second paragraph, 2.2.3 Movement annotations: Description of labeling snippets as “FM+”, FM-“. Please provide details about where FM+/- and FM++ belongs into these categories and provide the same in legend Figure 1.

Reply: While we agree this to be an important issue, paragraph 2 refers to already published data. We would thus like to refrain from introducing a more complex classification scheme for the temporal organization of FMs, because here we refer to 5s data units only. For future studies utilizing longer video sequences FM+/- and FM++ categories should definitely be taken into consideration.

4) First paragraph, 2.2.1 Participants: Description of infant inclusion criteria. Please use the term Typically Developing (TD) infants when listing up details.

Reply: We have added this as suggested.

5) First and second paragraph, 2.2.2 Multi-modal movement recordings: In first paragraph the authors use “... days post term age”, in second paragraph “...weeks post term age”. Please be consistent.

Reply: Thank you for spotting this. We consistently use weeks post term age except for describing the timing of the seven sessions, as we allowed for +/- 2 days for each of the seven data acquisition timepoints. For this occasion, we believe it is easier to comprehend for the reader to use “days” and have changed “post-term” in this sentence to “Time-points”.

6) Last sentences, 2.2.2 Multi-modal movement recordings: The authors describe a customized body-suit with designated pockets for IMU sensors. Customized for each individual infant? Was the body-suit customized in such a way that the IMU sensors did not move in relation of the infant body? Please clarify.

Reply: We have added a sentence on choosing age-specific suits according to infant size in paragraph 3 of this section.

7) Second paragraph, 2.2.3 Movement annotations: In the brackets, the authors define the labelling of “not assessable”. Please provide details about how “over-excited” is defined.

Reply: Thank you for pointing this out. We changed over-excited to “exhibiting a pleasure burst”.

Discussion:

8) As far as I can understand, the performance of the different sensor fusion modalities range between 82.1% and 90.7% using a single modality and between 90.7% and 93.4% using several modalities. Please provide clearer somewhere in the discussion – short - what clinical implications the findings might have, if any?, and how this relates to the clinical context that is described in the introduction (related to comment 1)). Maybe put it in somewhere in relation to the descriptions in last part of paragraph 4 in the Discussion?

Reply: Thank you very much for this suggestion. We wrote in paragraph 4 in the Discussion: “If different dimensions of input (from diverse modalities) for the same phenomenon (e.g., motion) are augmented, the representation of the ground truth (e.g., distinguishable patterns) may be improved, hence increasing the overall classification accuracy, as shown in our current study.” We also highlighted in Conclusion: “Complex evolving neurofunctions require equally complex (and very likely multi-sensor) assessments enabling a deeper understanding of infant development.” Scaling-up is in addition outlined in the fifth paragraph of the Discussion. We hope this fulfils the reviewers’ expectation.

9) Related to comment 8) about implications for clinical practice. In paragraph 5 the authors claim that an AI tool used in real-time clinical practice, needs to go beyond providing dichotomous labels (FM+ or FM-). Why? Today, expert-based GMA is used in such a way for prediction of CP. Why is this not the case for a possible AI tool? Please elaborate or clarify a bit if possible.

Reply: Thank you for pointing this out. We agree with the reviewer. To address this without engaging in a lengthy discussion (potentially confusing non-GMA expert readers) we have added “not only for detecting cerebral palsy” as – in line with a current global automated GMA working group – AI-tools for GMA need to look beyond the dichotomy of CP vs. no-CP and also include cohorts at elevated likelihood for various neurodevelopmental conditions. We would be happy if the reviewer agrees.

Reviewer #2 (Remarks to the Author)

Introduction: The authors effectively identify the 'gold standard' of General Movements Assessment (GMA) and highlight the superiority of assessments conducted by trained human assessors, as well as the limitations that currently impede the adoption of AI-driven GMA. The introduction of a sensor fusion approach using Inertial Measurement Units (IMUs) is innovative and successfully engages the reader with the work. The writing is accessible to a broad audience, well-referenced, and the research objectives are clearly articulated.

Reply: Thank you very much indeed for your positive evaluation and constructive comments which we addressed point-by-point below.

However, the introduction does not sufficiently address the rationale for developing an AI solution when trained human analysis is considered superior. I recommend that the authors include a discussion on the limitations of relying solely on trained human assessors and

assessments. This addition would clarify the necessity of an AI solution in addressing broader population health challenges. In the current state, the argument for AI's role, particularly within the infant provider community, lacks the compelling urgency required to challenge established care principles that have persisted for decades.

Reply: Thank you for this hint. We have amended the introduction accordingly to meet this need.

Methods: The methods section appropriately considers the time horizon of fidgety movement emergence and the optimal detection period. However, there is no information on the qualifications and experience of the two assessors, which is important for understanding the reliability of the human assessment component, especially since the IRR deteriorated upon a second analysis in some cases.

Reply: We have added the experience level of the two annotators in the Methods section, both are senior GMA experts having continuous clinical practices and being involved in more than 50 publications on GMA over the last decades.

Additionally, the use of a handheld HD camcorder raises questions, as a smartphone could achieve similar resolution while offering greater applicability in underserved areas. Providing justification of this selection is recommended.

Reply: Thank you for pointing this out. The HD camcorder, as a conventional device for GMA, was mounted on an aluminium-frame affixed to the cot. We added this information to the Methods. We fully agree with the reviewer and added a note that the HD camcorder could be substituted by a smartphone nowadays.

In section 2.4, "Neural Network Architectures," there is a typo in the first statement ("were we classified"), which should be corrected.

Reply: We have corrected this. Thank you for spotting this out.

The overall methodology is clear, well-described, and replicable. The accompanying graphics and tables contribute meaningfully to the understanding of the research.

Reply: Thank you very much.

Results: The results are clearly presented, well-supported by the data, and the analysis is robust and thorough. I have no further suggestions for this section.

Reply: Thank you.

Discussion: The discussion provides insightful and comprehensive analysis of the findings and proposes directions for future work. I suggest minimally expanding on the adoption issues impacting each modality (or modality combination). This would not only inform future research and technical innovation but also emphasize accessibility for a broader range of systems and populations that could benefit which is primary to the aims and scope of this publication.

Reply:
Thank you for this comment. We have discussed on the adoption of each modality, strengths and weaknesses, such as the non-intrusiveness, in the Discussion. In addition, we are referencing our recent publication in Communications Medicine where we elaborated on pros and cons of different sensor modalities.

Overall Recommendation: I consider this manuscript to be a significant contribution to the

literature on AI, GMA, and infant movement classification. I support its publication with minor revisions as outlined above. GMA is a critical component in clinical pathways, especially in neonatal intensive care units (NICUs), yet access to and training of expert providers remain barriers to widespread adoption. AI presents a promising and innovative strategy to overcome these challenges.

Reply: Thank you very much for your positive evaluation.

Reviewed by: Teresa Fair-Field, OTD, OTR/L (GMA-certified occupational therapist and infant movement classification researcher specializing in machine learning techniques)

Reviewer #3 (Remarks to the Author)

The study titled “Deep learning empowered sensor fusion boosts infant movement classification ” explores the application of AI - and specifically deep learning - in enhancing diagnostic processes using the Pechtl general movement assessment (GMA) to detect neurological impairments in infants. It specifically investigates a sensor fusion approach using pressure, inertial, and vision sensors to assess infant movements. The study compares various sensor combinations and fusion techniques (late and early fusion) to determine if a multi-sensor system can outperform single-sensor modalities. The findings reveal that the three-sensor fusion achieves a significantly higher classification accuracy (94.5%).

The paper is well written and is relevant from a scientific and clinical point of view. I have some concerns that I report below.

Reply: We would like to thank the reviewer for the positive review and the suggestions which we addressed point-by-point below.

It is not clear from the abstract what classification method has been implemented. Also, the first sentence in the abstract seems disconnected and uninformative from the rest of the text so I suggest removing it.

Reply: We have clarified which classification method was used (i.e., convolutional neural networks [CNN]). We have removed the first sentence as suggested.

I suggest concluding the introduction with a bulleted list of the innovative contributions of this work.

Reply: We have concluded introduction section with a bulleted list as suggested.

To improve readability the state of the art should occupy a single Section in which for each approach mentioned the limitations are analysed. Literature in the field of decision support systems based on deep learning should be up-to-date (i.e., 2021-2022-2023-2024).

Reply: We agree with the reviewer that having a separate section for the state of the art would improve readability. However, we must adhere to the formatting style of the journal which does not allow subheadings (subsections). We have re-checked that state-of-the-art literature on deep learning methods for automated movement assessment is up-to-date (2020-2024). Given the scope of this article, we refrain from analysing and comparing limitations of these studies, since most of them focus on movement classification based on one sensor modality. As there is no publicly available multi-sensory dataset, analysis of different sensor modalities was not possible in previous studies.

Authors are encouraged to share codes and data for the reproducibility of the experiments from other researchers.

Reply: We agree that sharing data and code is important for the reproducibility. We have stated in the manuscript, that dataset and code will be made publicly available upon acceptance (please see sections Data availability and Code availability).

What criteria will determine the release of the dataset? Will it be accessible to all researchers? Will the entire dataset be made available? I think that sharing this data with research groups in the field is crucial, as it represents a significant contribution of this work, especially since the proposed deep learning methodology itself does not introduce any innovations.

Reply: The dataset and code will be accessible to all researchers. We plan to publish our dataset and code at Zenodo (<https://zenodo.org/>). We will publish extracted 2D skeleton coordinates from the video data, pressure mat data, and IMU data.

Given the focus of the work, it would be important to have (1) a paragraph that well explains how multi-source data fusion occurs (the sensor fusion paragraph seems poorly informative), (2) how the input is secured to the architecture. All the blocks in Figure 3 should be explained in the text.

Reply: We now provide additional details on sensor fusion as suggested and explained all blocks in the Figure 3 in the text.

How was the data used synchronized?

Reply: Thank you for pointing this out. Data was synchronised by using time stamps for each sensor modality. Data were then aligned (synchronized) based on these time stamps. We have added this explanation in the body text.

It would be important to report a summary table of the training/testing and validation sets making it clear how much data from individual patients are in each of the sets (it is crucial to have a patient split for cross-validation purposes. Has this been implemented?).

Reply: We performed a 9-fold cross-validation where we split data into training and test sets. In each fold, we included 32 infants in the training set, and 4 infants in the test set. The training set was in addition split into two subsets where 7/8 [87.5%] of the training data was used for the parameter update, and 1/8 [12.5%] of the training data was used for the validation and early stopping to prevent overfitting. The number of snippets for each fold is presented in Table 1. We have now added statistics per infant for each fold: average number of snippets, standard deviation (SD) and range ([Min Max]).

I would avoid using the accuracy metric (Figure 4). Rather, I would plot and discuss the confusion matrices.

Reply: In Figure 4, we show balanced accuracy (BA), where $BA = (\text{Sensitivity} + \text{Specificity})/2$. The average sensitivity and specificity values (together with confidence intervals of mean) are presented in Table 2. Sensitivity (TPR – true positive rate) and specificity (TNR – true negative rate) are computed from confusion matrix, where FPR – false positive rate and FNR – false negative rate can be calculated by $FPR = 1 - TNR$, and $FNR = 1 - TPR$. Therefore, TPR, TNR, FPR, and FNR measures, represent confusion matrix.

In Figure 4, we were able to compare all methods in a concise way by using only one plot, where we show BA values for all methods and all nine folds together with mean BA across nine folds, and confidence intervals of mean (95%). To compare all methods using confusion matrices, one would need to show at least 8 confusion matrices, if average values across 9-folds would be shown (it is not possible to show error bars using confusion matrices, though), or $8 \times 9 = 72$ matrices if one wants to show data for all folds. Again, we provided Sensitivity and Specificity values in Table 2 which allows computing confusion matrices if needed. We believe that showing multiple confusion matrices would make comparison cumbersome without providing additional information. We sincerely hope the reviewer agrees.

We do very much hope the editors and reviewers are satisfied and happy with our changes and amendments made. We are indeed very grateful for the positive replies and feedback and hope the manuscript now fulfils the high standard for publication. Thank you very much for your time and efforts helping to amend our manuscript.

List of Changes

Dear Editor, dear Reviewers,

we sincerely thank you again for the constructive comments and valuable suggestions which have helped us to significantly improve our manuscript. We are very glad that our manuscript meets the reviewers' expectations and the high standards of your journal.

Thank you very much, indeed.

Yours sincerely,

Tomas Kulvicius on behalf of all authors

REVIEWERS' COMMENTS:

Reviewer #1 (Remarks to the Author):

I have reviewed all changes and ammendments made by the authors according to my comments and are satisfied and happy with all reponses. The manuscript now fulfils the standard needed for publication.

Reviewer #2 (Remarks to the Author):

Concerns have been satisfied.

Reviewer #3 (Remarks to the Author):

I thank the Authors for addressing my concerns, I have no further questions.